# Individual Nutrition Is Associated with Altered Gut Microbiome Composition for Adults with Food Insecurity

**DOI:** 10.3390/nu14163407

**Published:** 2022-08-19

**Authors:** Moira Bixby, Chris Gennings, Kristen M. C. Malecki, Ajay K. Sethi, Nasia Safdar, Paul E. Peppard, Shoshannah Eggers

**Affiliations:** 1Environmental Medicine and Public Health, Icahn School of Medicine at Mount Sinai, 1 Gustave L. Levy Place, New York, NY 10029, USA; 2Population Health Sciences, University of Wisconsin School of Medicine and Public Health, 610 Walnut St., WARF 707, Madison, WI 53726, USA; 3Department of Medicine, Division of Infectious Disease, School of Medicine and Public Health, University of Wisconsin-Madison, UW Med. Fndtn. Centennial Bldg., 1685 Highland Ave, Madison, WI 53705, USA; 4William S. Middleton Veterans Affairs Medical Center, 2500 Overlook Terrace, Madison, WI 53705, USA

**Keywords:** gut microbiome, mixture modeling, individual nutrition score, food insecurity

## Abstract

Diet is widely recognized as a key contributor to human gut microbiome composition and function. However, overall nutrition can be difficult to compare across a population with varying diets. Moreover, the role of food security in the relationship with overall nutrition and the gut microbiome is unclear. This study aims to investigate the association between personalized nutrition scores, variation in the adult gut microbiome, and modification by food insecurity. The data originate from the Survey of the Health of Wisconsin and the Wisconsin Microbiome Study. Individual nutrition scores were assessed using My Nutrition Index (MNI), calculated using data from food frequency questionnaires, and additional health history and demographic surveys. Food security and covariate data were measured through self-reported questionnaires. The gut microbiome was assessed using 16S amplicon sequencing of DNA extracted from stool samples. Associations, adjusted for confounding and interaction by food security, were estimated using Weighted Quantile Sum (WQS) regression models with Random Subset and Repeated Holdout extensions (WQS_RSRH_), with bacterial taxa used as components in the weighted index. Of 643 participants, the average MNI was 66.5 (SD = 31.9), and 22.8% of participants were food insecure. Increased MNI was significantly associated with altered gut microbial composition (β = 2.56, 95% CI = 0.52–4.61), with *Ruminococcus*, *Oscillospira*, and Blautia among the most heavily weighted of the 21 genera associated with the MNI score. In the stratified interaction WQS_RSRH_ models, the bacterial taxa most heavily weighted in the association with MNI differed by food security, but the level of association between MNI and the gut microbiome was not significantly different. More bacterial genera are important in the association with higher nutrition scores for people with food insecurity versus food security, including *Streptococcus*, *Parabacteroides* *Faecalibacterium*, and *Desulfovibrio.* Individual nutrition scores are associated with differences in adult gut microbiome composition. The bacterial taxa most associated with nutrition vary by level of food security. While further investigation is needed, results showed a higher nutrition score was associated with a wider range of bacterial taxa for food insecure vs. secure, suggesting nutritional quality in food insecure individuals is important in maintaining health and reducing disparities.

## 1. Introduction

The gut microbiome is increasingly being recognized as an important contributor to human health. The microbiome plays an important role in maintaining homeostasis of several key regulatory systems, including immune function and metabolism. An altered gut microbiome has been associated with a host of chronic conditions. Diet and exposure to antimicrobials are some of the best known modifiable factors to influence gut microbiome composition [1]. Opportunities for intervention via diet and microbiome have led to numerous investigations over the last decade. However, previous studies of diet and the gut microbiome primarily focus on specific dietary components (fiber [2], fat [3], etc.), or interventions using specific assigned diets (Mediterranean [4], etc.). Few studies have investigated the relationship between the overall nutrition of a person’s regular diet and composition of the gut microbiome and how social determinants such as poverty and food insecurity shape these relationships. Little is known about the relationship between food insecurity and the gut microbiome. Because food insecurity can affect the quality of food consumed [5], and influence biological processes related to stress [6,7], it is likely that food security would modify the effect of overall nutrition on the gut microbiome.

A more holistic assessment of dietary quality and associations with the gut is needed to reflect real-world dietary patterns and the impact of microbial composition as the gut responds to the entire diet, not simply specific components. Furthermore, it is difficult for people to drop all of their dietary habits and adopt a completely new diet, and maintain those changes over time, and thus, interventional studies have limited effectiveness in real world settings. An assessment of overall dietary quality could support more realistic interventions that can be implemented and sustained over time with greater effectiveness than previous interventions.

Diet and age are inter-related components shaping the gut microbiome in adult-hood. The gut microbiome primarily takes shape in the first few years of life, and remains relatively stable over time, but can shift dramatically with changes in dietary and other xenobiotic exposures [8,9,10]. The human gut is typically colonized by four main bacterial phyla, Actinobacteria, Bacteroidetes, Proteobacteria, and Firmicutes, with Bacteroidetes and Firmicutes typically the most dominant, and an overabundance of Proteobacteria signaling dysbiosis and epithelial dysfunction [11,12,13]. The evidence around agrarian diets, high in fiber, compared to Western diets, high in fat and low in fiber, is fairly robust [14]. Diets that have been conserved since the onset of human agriculture have very different gut microbiota compared to Western diets. In a landmark investigation of children from rural Africa compared to children from European cities, the African children with higher fiber diets had greater diversity of the gut microbiome and increased abundance of *Prevotella* and other bacteria genera capable of cellulose and xylan degradation and short-chain fatty acid production (SFCA), not abundant in the European gut microbiomes [8]. Animal and human studies of habitual diet and dietary interventions focused on fiber and fat consumption show a consistent increase in the ratio of *Prevotalla* to *Bacteroides* in high fiber vs. high animal fat diets, and generally higher abundance of SCFA-producing bacteria such as *Roseburia* and *Ruminococcus*, and lower abundance of mucus-degrading and bile-tolerant bacteria such as *Alistipes* and *Bilophila* [3,9,14,15,16].

While our understanding of specific dietary components within Westernized and agrarian diets, such as fiber and fat, and their contribution to gut microbial composition is fair, there are different kinds of Westernized diets with different nutritional value that may contribute differently to gut microbial composition. For instance, the composition of the gut microbiome in Western populations following vegan, vegetarian, omnivore, and Mediterranean diets are significantly different [4,17,18]. Nutritional guidelines set by the U.S. Department of Health and Human Services (USDHS) and the U.S. Department of Agriculture (USDA) vary from person to person based on age, height, sex, weight, and medical status (pregnancy, diabetes, etc.) [19]; thus, a diet that is considered nutritious for one person may not be for another. However, general nutrition of a diet (as determined by USDHS and USDA guidelines), to our knowledge, has not been considered in association with gut microbial composition.

An additional factor contributing to differences in diet for Western populations is food insecurity. Defined by the USDA, food insecurity occurs when “access to food is limited by lack of money or resources.” Food insecurity was prevalent in greater than 11% of households in the United States in 2018 [20], and has been significantly associated with sleep disorders, depression, and anxiety, although the etiological pathway is not completely clear [21]. Food insecurity is not always associated with a lack of energy consumption, but the quality of food items is often lower among individuals with food insecurity. Food insecure households often live in areas where access to high quality food is limited or more cost prohibitive. Food insecurity among American households is often chronic, therefore impacting long-term dietary intake, leading to both under and over nutrition [20]. Diets characterized by high caloric, high fat, high sugar, and often low nutrient-dense foods, including processed carbohydrates, are often less expensive and more common when food resources are scarce [22]. Moreover, more nutritious diets, such as the Mediterranean diet, are often not accessible for people experiencing food insecurity [23]. Food insecurity is conversely linked with undernutrition, defined as inadequate intake of essential nutrients and can occur minimally or have longer-term effects [22,24]. Over nutrition in the United States has been linked with obesity and altered Firmicutes to Bacteroides ratios, leading to obesity-related inflammation [22]. Undernutrition among children in Bangladesh has shown limited gut microbiome diversity and overgrowth of more harmful bacteria [25]. Few studies within the United States have examined the contributions of food insecurity on microbial diversity, particularly in adult populations.

The aim of this study was to assess the association between personalized nutrition scores, using My Nutrition Index (MNI), and the composition of the gut microbiome in adults, using 16 s rRNA amplicon sequencing. We further investigated whether food security modifies the association between MNI and the gut microbiome. To address these aims, we used data from the Survey of the Health of Wisconsin (SHOW) and the Wisconsin Microbiome Study (WMS). We applied Weighted Quantile Sum (WQS) regression techniques to the microbiome data analysis to investigate associations with the gut microbiome as a mixture, applying a stratified interaction WQS model to microbiome data, which has not been done previously.

## 2. Methods

### 2.1. Study Population

This analysis used existing data collected by the Survey of the Health of Wisconsin (SHOW), and its ancillary Wisconsin Microbiome Study (WMS). Descriptions of both studies have been previously published [26,27,28]. In brief, SHOW is a population-based health examination survey and ongoing cohort study that collects a wide range of survey data, objective measurements, and biological specimens relating to health exposures and outcomes, from non-institutionalized residents of Wisconsin. In 2016–2017, WMS added the collection of survey data on microbial exposures and risk factors, and bio-specimens for microbiome analysis for all adult participants. WMS participants contributed one stool microbiome sample each, with the survey data taken cross-sectionally. Exclusion criteria included having incomplete data on the variables of interest found in Table 1, including the components that comprised our primary outcome, the My Nutrition Index. For this analysis, we used survey self-report data, and microbiome sequencing data from stool samples, from 643 participants with complete data. 

The protocol for this analysis was approved by the Icahn School of Medicine at Mount Sinai Institutional Review Board (STUDY-20-01396, Approved 19 January 2021). SHOW guidelines for data sharing and publication have been followed.

### 2.2. Gut Microbiome Analysis

The stool collection, DNA extraction, and sequencing data processing have been previously described in detail [27,29]. Briefly, stool samples were collected and refrigerated at home up to 24 h prior to a study visit. Samples were then shipped on ice from the study clinic to the lab within 24 h, where they were aliquoted and stored at −80 °C until DNA extraction. Genomic DNA was extracted using chemical, heat, and mechanical lysis. DNA purification was performed with a phenol-chloroform-isoamyl alcohol extraction followed by a clean-up kit. DNA was quantified before PCR amplification of the 16 s rRNA V4 region. Custom barcoded PCR primers were used following the protocol by Kozich et al. [30]. Amplicon sequences were purified using a low-melt agarose gel electrophoresis, and further cleaned using a 96 well cleanup kit. Samples were quantified and pooled before sequencing on an Illumina MiSeq per manufacturer’s instructions. Raw sequencing data were processed in mothur v. 1.39 [30] following the MiSeq data Standard Operating Procedure [31]. Overlapping sequences were aligned using the SILVA v.132 16S rRNA reference database [32]. Low-quality reads and chimeras were removed using UCHIME [33]. Operational taxonomic units (OTUs) were assigned at the species level (97% similarity) using GreenGenes v. gg_13_8_99 [34]. OTU counts were rarified to 10,000 sequences per sample. Relative abundance was calculated for all OTUs with greater than 10% detect, and further ranked. We ranked the OTUs across subjects by setting zeros to a rank of 0, and deciling non-zero values.

### 2.3. My Nutrition Index

The My Nutrition Index (MNI) is a validated index of the nutritional value of one’s daily diet derived from self-reported food frequency questionnaires (FFQ) [35,36,37]. It comprises 34 dietary components: total fat, saturated fat, monounsaturated and polyunsaturated fat, energy, protein, carbohydrates, alcohol, caffeine, sugar, fiber, vitamin E as alpha-tocopherol, vitamin C, cholesterol, potassium, sodium, calcium, magnesium, iron, phosphorus, zinc, thiamin, riboflavin, niacin, vitamin B_5_, vitamin B_6_, vitamin B_12_, vitamin A, vitamin D, vitamin K, manganese, chloride, folate, and selenium. The FFQ used to collect dietary nutrient values was the Diet History Questionnaire II for US & Canada, produced by the National Cancer Institute [38]. Usual diet over the previous year was queried using the FFQ, and responses were processed into nutrient values using the Diet*Calc software [39].

MNI is a metric of how close each component is to guideline values based on the characteristics of the subject (for example, body mass index, smoking status (yes/no), pregnancy, sex, age, and physical activity). It assigns higher scores for nutrient values that fall within the published dietary guidelines’ recommended concentration range, and lower scores if intake for a given nutrient deviates from this ideal range (above or below). MNI provides an overall index score ranging from 0 to 100, with higher scores reflecting a more nutritious diet. Four subscales focus the assessment to nutritional categories: Vitamin Index, Mineral Index, Electrolyte Index, and Macro Nutrient Index. Each is on the same scale as the MNI. See the Appendix A for further description of the subscales. Values for the MNI and subscales above roughly 90 indicate adequate nutrition on each scale. In preprocessing the nutrient data, extreme values (>99th percentile) of sodium and energy were removed from the analysis. The subscales were evaluated as dichotomized values due to skewed and uniform distributions. The Macro Index, Vitamin Index, and Mineral Index were split at a score of 90 due to skewness; the Electrolyte index (here, comprised of sodium and potassium) was split at the median due to a uniform distribution. 

### 2.4. Covariates

Due to the use of sex, age, BMI, smoking, and physical activity in the calculation of MNI, we excluded those variables as covariates or confounders in our analyses. Therefore, our only a priori confounder included self-reported antibiotic use in the past year (yes vs. no/missing). Additional covariates considered included education level, race, family income, and food security. Food insecurity was considered a confounder in stratified analyses, and as an effect modifier in interaction analyses. Food security status was determined by pooling responses to the following three questions: (1) In the last 12 months, did you ever get emergency food from a church, a food pantry, or a food bank, or eat in a soup kitchen? (2) In the last 12 months, have you been concerned about having enough food for you or your family? (3) In the last 12 months, were you authorized to receive Food Stamps, which includes a food stamp card or voucher, or cash grants from the state for food? If the subject responded yes to any of these questions, they were considered to be food insecure. Additional description of other covariate measurement is included in the Appendix A.

## 3. Statistical Analyses

Statistical analyses were performed in R version 4.0.4 R Core Team (2020), and SAS 9.4 (SAS Institute, Cary, NC, USA). We analyzed the microbiome as a mixture in an association model with each nutritional index, our primary analysis being with MNI, and further stratified the analyses by food security level. The primary mixture method employed was weighted quantile sum regression (WQS) with random subsets (WQS_RS_) and repeated holdouts (WQS_RH_) extensions (WQS_RSRH_), with secondary analyses using stratified interaction WQS_RSRH_. Additional details about the model building approach have been included in the Appendix A.

The WQS_RS_ and WQS_RH_ have been previously described [40,41]. Briefly, the WQS_RS_ is an appropriate analysis for when the number of predictors exceeds the number of subjects, as is the case in this study. The predictor variables, in this case the OTUs, are ranked, here as 0s and deciled non-zeros. Using these ranks, weights were calculated for each OTU in the mixture in association with the outcome of interest and adjusted for covariates. The calculation of the weights was estimated 1000 times, with each estimation using a random subset of the OTUs, wherein 23 OTUs were randomly selected to estimate the weights for each random subset analysis; the final WQS weights were calculated as a weighted average. The resultant WQS index for each participant is the sum of the product of each subject’s OTU rank*weight within the mixture. This index is then applied in a generalized linear model using the WQS as input, and adjusting for covariates, in association with the outcome. The datasets were split into 40/60 training/validation datasets to enhance generalizability, where the weights were estimated on the training and tested on the validation dataset in the generalized linear model. Statistical significance was determined with *p* < 0.05.

The repeated holdout extension (WQS_RH_) of this extends the number of times the weights are estimated such that 1 WQS_RS_ analysis is performed a determined number of times, and the average weights across the repeated holdouts is taken as the final weights used in the generalized linear model to test on the validation dataset. We performed repeated holdout analyses 100 times for primary analyses. 

In WQS, a stratified analysis estimates the weights within the stratified groups, such that separate weights were estimated for each of the OTUs within the food secure and food insecure groups. The WQS interaction analyses tests for different levels of association between the two food security groups; however, the weights of the OTUs within the mixture would not differ [42]. The stratified interaction analysis estimates different association levels between the microbiome mixture, in addition to different weights in association to the specific outcome, between the food secure and insecure groups. 

These various WQS methods output weighted indices that assign a weight to each individual OTU within the mixture. Were each distinct OTU to have an equal contribution to the association with the outcome, each OTU would have the same weight value. Assuming the OTUs do not have equal weight (equi-weight), we calculate a threshold of 1/c, c being equal to the total number of components in the mixture. If the OTUs are not equi-weighted, we are interested in the weights that are greater than the threshold because they have a greater impact on the positive association between the mixture and the MNI. Instead of assessing the weights at the OTU level, however, we summed the weights of the OTUs within each genus, for a genus-level weight. The threshold, then, became 1/89, since there were 89 genera. 

## 4. Results

### 4.1. Study Sample

Table 1 provides characteristics and demographics of the analytic sample from SHOW. The study sample (*n* = 643) consisted of adults (average age 57 years), who primarily identified as white (87%). The average body mass index (BMI) was obese, at 30.6. The majority of participants were female (57%) and had a high school degree or higher. The majority of participants (78%) were food secure (yes/no), and reported having never smoked (87%). In total, 60% of participants had not taken antibiotics in the year prior to providing their stool sample. The microbiome samples (1 per participant) had a total of 6645 operational taxonomic units (OTUs). After filtering the analysis to OTUs that were detected in 10% or more of the samples, there were 516 OTUs comprising 135 species, 89 genera, and 8 phyla. Covariates for the My Nutrition Index included race, food insecurity, and education status; the binary electrolyte index was associated with food insecurity and education. The macro index was associated with education. The mineral index was not associated with any covariates. The vitamin index was associated with antibiotic use in the past year. See Appendix A for further details. 

### 4.2. Associations between My Nutrition Index and the Gut Microbiome

We found a positive association between the microbiome mixture and the MNI (Table 2, Appendix A). The positive association suggested that for each non-zero decile increase in the WQS microbiome index, the MNI increased by 2.56 units (β = 2.56, CI = 0.52, 4.61). This indicates that, when there is a 1 out of 10 increase in the WQS index, due to a combination of (a) higher rank of one or more OTU, indicating greater exposure, and/or (b) exposure to OTUs that are weighted more highly in a positive association to the MNI, the MNI is increased (100 = top health, 0 is lowest health). 

In total, 21/89 genera had sums above the genera-specific threshold from the repeated holdout. Moreover, 15 of the 21 genera were within the Firmicutes phylum, with unclassified *Lachnospiraceae* as the genera weight with the highest summed weight, and unclassified *Ruminococcaceae* as the second highest summed weight. The phyla Bacteroidetes, Proteobacteria, and Verrucomicrobiota also ranked highly from this analysis (see Figure 1 and Appendix A). To view the magnitude of the weights, see Appendix A. 

### 4.3. Associations between Nutritional Subscales and the Gut Microbiome 

We found a positive association between the microbiome WQS mixture and the binary electrolyte index, which suggested that for each decile increase in the WQS_RS_ microbiome index, there was a 58% increase in the odds of having an above median electrolyte index score versus a below-median score (OR = 1.58, CI = 1.24, 2.02) (Table 2). 

In total, 17/89 genera had sums above the genera-specific threshold from the repeated holdout. Moreover, 14 of the 17 genera were within the Firmicutes phylum, with unclassified *Lachnospiraceae* as the genera weight with the highest summed weight, and unclassified *Ruminococcaceae* as the second highest summed weight (similar to the MNI analysis). The phyla Proteobacteria and unclassified Bacteria also ranked highly from this analysis (see Figure 1 and Appendix A). 

### 4.4. Associations Stratified by Food Insecurity

#### 4.4.1. My Nutrition Index

We found a significant positive association between the WQS microbiome index and the MNI when stratified by food security status and adjusted for covariates (β = 7.7, CI = 1.32, 14.1) (Table 3). In the stratified analysis, the difference in strata is calculated in the estimation of the weights; therefore, the interpretation remains the same for both food secure and insecure groups: there is a positive association between the gut microbiome and the MNI. However, the OTUs that contribute to the positive association differ between the food secure vs insecure groups. See Figure 1 and Appendix A for the stratified weights above the stratum-and-genera-level threshold. Instead of reporting within strata the OTU weights categorized by the genera, we summed the weights of OTUs within strata and genera. We then calculated the stratum-specific threshold to compare genera that were shown to have greater influence on the mixture than if each genera had an equal weight. With 39 genera total and 2 strata, 1/78 was the threshold value in this analysis. We found 25 genera to be above the threshold, all of which were identified in the food insecure group as highly weighted genera, while 12 of these 25 were also identified as highly weighted in the food secure group (see Figure 1, Appendix A). 

The highest weights within this analysis were the unclassified *Lachnospiraceae* and the unclassified *Ruminococcaceae*, which were weighted the highest in both the food secure and insecure groups; however, the weights were higher for each genus in the food insecure group than in the food secure group. Genera whose summed weights were above the genera-and-stratum-specific weights and that were exclusively above the threshold in the food insecure group included *Lactobacillus*. 

There was no significant association with an interaction between food security status and the WQS index on the MNI, with or without stratification by food security status (see Appendix A).

#### 4.4.2. Nutritional Subscales

We found a significant positive association between the WQS microbiome index and the electrolyte index when stratified by food security status and adjusted for covariates (β = 2.86, CI = 1.53, 5.37) (Table 3). There is a positive association between the gut microbiome and the electrolyte index; however, the OTUs that contribute to the positive association differ between the food secure vs insecure groups. See Figure 1 and Appendix A for the stratified weights above the stratum-and-genera-level threshold. We found 26 genera to be above the threshold, all of which were identified as highly weighted in the food insecure group, while only 14 of these 26 were also identified in the food secure group. 

The highest weights within this analysis were the unclassified *Lachnospiraceae* and the unclassified *Ruminococcaceae*, for both the food secure and insecure groups. Similar to the MNI analysis, the weights for these taxa were higher in the food insecure group than in the food secure group. Genera whose summed weights were above the genera-and-stratum-specific weights and that were exclusively above the threshold in the food insecure group included *Peptoniphilus*, *Coprabacillus*, *Actinomyces*, *Lactobacillus*, *Anaerostipes*, *Ruminococcus*, *Faecalibacterium*, *Desulfovibrio*, *Akkermansia*, and *Streptococcus*. No significant interactions with food security were seen in any of the subscale analyses (see Appendix A).

## 5. Discussion and Conclusions

This analysis is among the first to consider total dietary quality measured by personalized nutrition scores (MNI) and specific indices on gut microbial composition, and uses a novel approach to microbiome analysis with stratified weighted quantile sum regression. The application of this WQS approach to account for the mixtures of both dietary components and gut microbial composition acknowledges the complexity of both dietary as well as gut microbial composition and overcomes limitations of previous analyses of diet and gut microbiome. This analysis also extended the application of WQS to microbiome data, as it is the first analysis to apply the stratified interaction WQS extension to microbiome data. 

Using this novel approach, we found a significant association between the adult gut microbiota and personalized nutrition scores (MNI) and the electrolyte index. There were significantly different gut microbes associated with improved nutrition scores for people with food security vs. food insecurity, but no evidence for significant change in level of overall association (i.e., different slope). More bacterial genera were identified as important in the association with higher nutrition scores for people reporting food insecurity. This may suggest nutrition is more important in shaping the gut microbiome for people who are food insecure, but this requires additional exploration. Food insecurity has been associated with numerous adverse metabolic and immune dysregulation and is often an overshadowed social determinant of health. Findings point to a potential mechanism by which the gut microbiome may play a role in exacerbating persistent disparities in numerous health outcomes, including cardiometabolic health. Findings from this analysis suggest nutrition is more important in shaping the gut microbiome for people who are food insecure, but this requires additional exploration.

A greater number of Firmicutes than any other phylum was associated with improved MNI and electrolyte scores, including most heavily weighted *Oscillospira* and *Ruminococcus*, as well as unclassified *Ruminococcaceae* and *Lachnospiraceae*. In an analysis of the Mediterranean diet and body weight in a small sample of Spanish adults, *Oscillospira*, *Desulfovibrio* and *Christensenellaceae*, which were all heavily weighted in our indices, were all enriched in the normal weight group compared to the overweight group [43]. *Christensenellaceae* was also associated with better compliance to the Mediterranean diet [43,44], and *Oscillospira* were enriched in those who ate less protein and cholesterol [43]. *Oscillospira* may be involved in the digestion and absorption of carbohydrates that are not digestible by humans [45,46,47], and have been shown to be inversely associated with inflammatory conditions such as inflammatory bowel and Crohn’s disease [48], and positively associated with leanness in several different populations [45,47,49,50,51]. *Christensenellaceae* have also been associated with leanness [51] and inversely associated with inflammation, atherosclerotic plaque formation [52], and Parkinson’s disease [53]. *Streptococcus*, which is weighted above the threshold in the MNI and electrolyte indices, has been associated with high BMI, high fat intake, and low adherence to the Mediterranean diet [43,54], and has been linked to colorectal cancer, inflammation, and atherosclerotic disease [54,55,56]. This may seem contradictory to the other findings; however, it is important to note that a high proportion of our study population is overweight or obese, and thus, it follows that *Streptococcus* would be enriched even with a high nutrition score. Moreover, while high meat consumption is not always considered nutritious, meat, particularly processed meat, is high in electrolytes (sodium and potassium), which may be helping participants reach the dietary guidelines for those nutrients, increasing their MNI and electrolyte scores. Considered together, our findings are consistent with these other studies of nutrition and health.

Several genera were significantly associated with both the MNI index and the electrolyte index, suggesting that dietary electrolytes may be driving the association we see with MNI overall. *Oscillospira*, *Ruminococcaceae*, and *Christensenellaceae* have all been associated with high salt diets in studies of mice and rats [57,58,59]. It has been hypothesized that changes in gut microbial composition help absorption of salt into the body and contribute to the etiological pathway between salt consumption and hypertension for people with salt sensitivity [60]. In a multipart study of salt consumption, microbiome composition, and hypertension in both mice and human participants, Wilck et al. found that *Lactobacillus* species were depleted, blood pressure increased, and T helper 17 (T_H_17) cells, which are thought to contribute to hypertension, increased with higher salt consumption. In mice fed a high salt diet, supplementation of *Lactobacillus* species with a high salt diet reduced T_H_17 abundance and ameliorated salt-sensitive hypertension [61]. An observational study in a large Chinese cohort found dietary potassium density to be associated with many gut microbial taxa, including *Dorea*, *Oscillospira*, *Lachnospiraceae*, and *Ruminococcus* [62], which were also heavily weighted in association with both the MNI and electrolyte indices. These findings along with the findings of heavily weighted *Oscillospira*, *Ruminococcaceae*, and *Christensenellaceae*, as well as *Lactobacillus* not being heavily weighted in the positive WQS_RSRH_ index with either the electrolyte or MNI index, are consistent with previous studies. However, it is important to note that salt consumption that is too low and too high both contribute to lower MNI and electrolyte index scores. Considering our findings in this context, it is clear that electrolyte consumption, salt in particular, is associated with distinct shifts in gut microbial composition.

In the analyses stratified by food insecurity, we found that all the genera that were heavily weighted for the food secure group were more heavily weighted for the food insecure group than for the food secure group. The food insecure group was also associated with 13 genera that were not associated with the food secure group across both the MNI and electrolyte indices, with 9 identified in both indices. Although the weighted indices were different for the food secure and insecure groups, the slope of association between the groups was not statistically different from each other; however, the beta estimates for the WQS indices in the stratified analyses were approximately twice the estimates in the unstratified analyses. We suspect that these increased estimates are likely due to a reduction of noise, and thus, a reduction in bias towards the null, once the indices were specified to the two food security groups.

Differences in WQS weights for people who are food secure and food insecure may be driven by differences in dietary components making up the MNI score. Because the MNI uses dietary nutrient data extracted from the FFQ and is calculated specifically to each individual’s recommended nutrition guidelines, individuals with very different dietary components can have the same MNI score. For instance, salt and fat are both higher in lower cost diets [63], and people who are food insecure consume fewer fruits, vegetables, and dairy products than people who are food secure [64]; thus, it is likely that dietary components leading to the same MNI score are different for people who are food secure than people who are food insecure. 

Chronic stress related to lack of food and other factors associated with food insecurity may also contribute to differences in gut microbiome composition for food insecure people. Stress and gut health are linked through the gut–brain axis, and people experiencing chronic stress are more susceptible to gut diseases [65]. Stress has been shown to increase the abundance of *Clostridium*, *Oscillibacter*, *Anaerotruncus*, and *Peptococcus*, and decrease the abundance of *Lactobacillus*, *Bacteroides*, and *Porphyromonadaceae* in mammalian gut microbiomes [66,67,68,69,70]. In our stratified analysis, *Clostridium* and *Bacteroides* were more heavily weighted for the food insecure group in the association with MNI and the electrolyte indices. Moreover, *Lactobacillus*, which was not associated with either index in the unstratified analysis, was associated with both indices for the food insecure group only in stratified analysis. In the case of *Lactobacillus*, and to some extent *Bacteroides*, high stress and potentially high salt intake in the food insecure group may result in low average abundance of these bacteria; however, better nutrition may ameliorate the lack of abundance, resulting in heavy weights in the WQS indices. 

Alternative explanations for differences in the gut microbiome with nutrition and food insecurity may include differences in chemical contamination, and in the microbial response to nutrients. Because people experiencing food insecurity eat more highly processed foods, those processing steps may introduce chemical additives as well as chemical contamination from equipment and packaging. While these chemical exposures are likely at low levels, they may be enough to suppress the growth of some bacteria [71]. Moreover, introducing more nutritious and less processed foods to the diet may induce a rapid amount of growth, both by diluting the concentration of chemical contaminants and by adding previously unavailable nutrients. While our findings contribute additional information on food security and the gut microbiome in an adult population, further investigation is needed to understand the biological mechanisms. 

We have previously demonstrated the use of WQS methods with microbiome data [72]; however, this is the first analysis to apply the extended WQS stratification and interaction models to microbiome data. This analysis also adjusted the preprocessing steps to include deciles above 0 in the ranking of the WQS indices, instead of tertiles above 0 as previously demonstrated. The ability of the models to identify an overall association and specific taxa consistent with previously published analysis can be considered further validation of the use of WQS with microbiome data. 

The MNI is a relatively new tool for dietary evaluation. We chose to use MNI in this analysis instead of other more widely used and validated measures, such as the Healthy Eating Index (HEI) [73,74], because the MNI has a few key advantages. Unlike the HEI, MNI can adjust for characteristics that may alter an individual’s recommended nutritional guidelines. The MNI also accounts for excess micronutrient intake that may lead to adverse health effects. In a previous comparison of MNI to HEI using data from the National Health and Nutrition Examination Survey (NHANES), the HEI and MNI showed similar associations with several health outcomes [37]. The two measures were mildly correlated; however, MNI had more variation. The HEI and MNI showed a curvilinear relationship, in that values were more similar above the halfway point of each scale, suggesting that MNI is more sensitive to poor nutrition. Measurement of poor nutrition was particularly important in this analysis, as it is associated with food insecurity. 

While this study contributes new information to the field, it also includes some limitations. While we think of gut microbiome composition as being a result of diet, the WQS equation is built to have the mixture index on the predictor side of the equation. However, because data are cross-sectional, we are restricted to examining associations and not causality, and thus, the directionality of the equation is negligible in this analysis. We also assume that the single timepoint of gut microbiome analysis is representative of the usual composition of the gut microbiome. This analysis is also limited by the accuracy of dietary recall for nutritional information; however, the FFQ is a validated instrument frequently used in similar studies of dietary habits. It is worth noting, however, that the accuracy of dietary recall may be different for people with and without food insecurity. Food frequency questionnaires have been shown to be less accurate for people with low SES, and low education level, which are both correlated with food insecurity [75,76,77]. 

The use of covariates in our analysis were limited due to the inclusion of many demographic characteristics in the calculation of the nutritional indices. However, we chose to include covariates, including self-reported race, when statistically significant in order to reduce residual confounding. The inclusion of race in our models does not suggest that there are biological differences by race in the pathways between nutrition, food security, and gut microbiome composition. More likely, there are social and structural mechanisms, including racism, associated with race that influence food security level, nutritional exposures, and other potential exposures that may influence gut microbial composition [78]. While the use of race as a covariate in this study is not ideal, the measurement of all potentially confounding covariates associated with race was not feasible for this analysis. Furthermore, by demonstrating the importance of food security in the pathway between nutrition and the gut microbiome, we hope to encourage other microbiome researchers to include variables such as food security in their analysis rather than relying on race in their analyses as a ghost variable, encompassing all social and structural drivers of racial differences in the microbiome [79].

Future directions from this analysis include longitudinal studies with repeated gut microbiome sample collection and 24 h dietary recall corresponding to each sample. This longitudinal data structure would allow for investigation of the relationship between MNI, food security, and the gut microbiome over time, with more precisely timed measurement of nutrition and the gut microbiome, as well extending the use of lagged WQS models to longitudinal microbiome data.

In conclusion, individual nutrition scores are associated with the adult gut microbiome, and the bacterial taxa most associated with nutrition vary by level of food security. Results may suggest that better nutrition is more important in shaping the gut microbiome for people who are food insecure, because nutrition was associated with a wider range of bacterial taxa for food insecure vs. secure individuals. However, additional research is needed.

## Figures and Tables

**Figure 1 nutrients-14-03407-f001:**
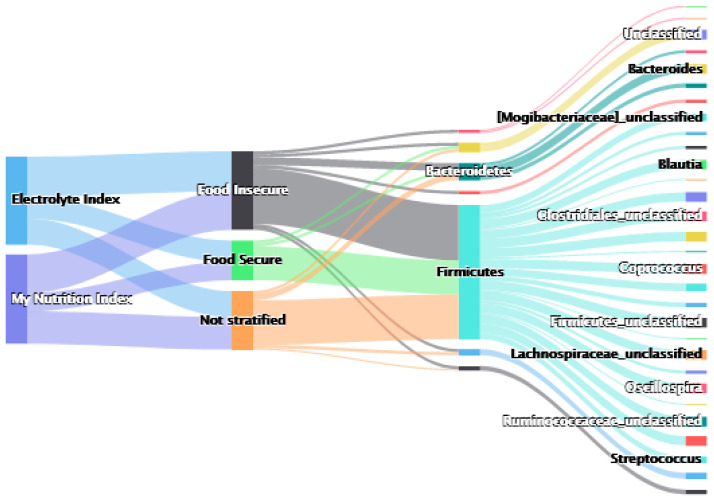
Sankey Plot illustrating the overlap of genera with weights above the threshold (summed OTU weights within genera) from each of the six statistically significant WQS_RSRH_ analyses testing for association between the microbiome mixture and the nutritional indices. The genera represented show that each genus was found to be highly weighted. Nodes, from right to left, display the WQS_RSRH_ analysis outcome (My Nutrition Index/Electrolyte Index), stratification of WQS_RSRH_ weights by food security (not stratified/food secure/food insecure), bacterial phylum, and bacterial genus. The widths of links between nodes are proportional to the number of genera in common to each set of nodes. Scrolling over the links between nodes displays the number of genera found to be above the weight threshold in the WQS_RSRH_ analyses, dependent upon the relationship between the nodes. Click on the link to see the GIF: https://rpubs.com/bixbym/933453.

**Table 1 nutrients-14-03407-t001:** Characteristics and demographics of subjects stratified by food security status.

Characteristic	N	Food Secure ^1^, N = 489	Food Insecure ^1^, N = 144	*p*-Value ^2^
My Nutrition Index	633	62.5 (20.4)	47.3 (25.6)	<0.001
Age	633	56.4 (16.0)	49.3 (15.6)	<0.001
Alcohol consumption (g/day)	633	10.4 (28.0)	15.4 (61.2)	0.3
Body Mass Index	624	29.8 (7.0)	33.4 (9.0)	<0.001
Poverty to Income Ratio	615	4.5 (2.8)	1.6 (1.0)	<0.001
Gender	633			0.2
Male		213 (44%)	54 (38%)	
Female		276 (56%)	90 (62%)	
Race	632			<0.001
White		431 (88%)	90 (62%)	
Other/Non-White		57 (12%)	54 (38%)	
Antibiotic use in the Past Year	633			0.2
Did not Use		303 (62%)	80 (56%)	
Did Use		161 (33%)	52 (36%)	
Unknown/Missing		25 (5%)	12 (8%)	
Education	633			<0.001
<High School		20 (4%)	19 (13%)	
High School or Associate’s Degree		251 (51%)	103 (72%)	
Bachelor’s Degree or Higher		218 (45%)	22 (15%)	
Smoking Status	619			<0.001
Never		305 (64%)	60 (42%)	
Current		37 (8%)	43 (30%)	
Former		134 (28%)	40 (28%)	
Electrolyte Index	633			<0.001
<Median		225 (46%)	93 (65%)	
≥Median		264 (54%)	51 (35%)	
Vitamin Index	633			0.2
<90		438 (90%)	134 (93%)	
≥90		51 (10%)	10 (7%)	
Macro Nutrient Index	624			0.015
<90		313 (65%)	105 (76%)	
≥90		172 (35%)	34 (24%)	
Mineral Index	633			0.11
<90		224 (46%)	77 (53%)	
≥90		265 (54%)	67 (47%)	
Shannon Diversity Index	624	3.3 (0.5)	3.1 (0.5)	<0.001
Diabetes (Type 1 or 2)	568			0.002
Yes		52 (12%)	30 (23%)	
No		384 (88%)	102 (77%)	
Chronic Conditions	633			<0.001
Yes		210 (43%)	91 (63%)	
No		279 (57%)	53 (37%)	

^1^ Mean (SD); *n* (%); ^2^ Welch two-sample *t*-test; Pearson’s Chi-squared test.

**Table 2 nutrients-14-03407-t002:** Results from the repeated holdout WQSRS regression, with positively constrained betas, for the Gaussian and logit models with non-zero OTUs deciled in generalized linear models and adjusted for covariates. MNI *n* = 623, Electrolyte index *n* = 624.

	MNI	Electrolyte Index
	β (95% CI)	OR (95% CI)
(Intercept)	47.9 (38.4, 57.4)	0.15 (0.06, 0.37)
WQS	**2.56 (0.52, 4.61)**	**1.58 (1.24, 2.02)**
Antibiotic use in past year: Yes (vs. no)	0.58 (−2.38, 3.54)	1.19 (0.89, 1.60)
Antibiotic use in past year: Unknown (vs. no)	−2.42 (−10.73, 5.89)	1.09 (0.58, 2.04)
Education: High school/associate’s degree (vs. less than high school degree)	9.0 (0.77, 17.23)	3.13 (1.45, 6.75)
Education: Bachelor’s degree or higher (vs. less than high school degree)	13.5 (4.88, 22.24)	3.94 (1.74, 8.94)
Race (non-white vs. white)	−0.1 (−13.2, −4.4)	*NA*
Food insecurity (insecure vs. secure)	−10.02 (−13.85, −6.2)	0.61 (0.45, 0.83)

**Table 3 nutrients-14-03407-t003:** Results from the stratified repeated holdout WQSRS regression, with positively constrained betas, for the Gaussian and logit models with non-zero OTUs deciled in generalized linear models and adjusted for covariates, stratified by food security status (food secure versus insecure). MNI *n* = 623, Electrolyte index *n* = 624.

	MNI	Electrolyte Index
	β (95% CI)	OR (95% CI)
(Intercept)	49.8 (41, 58.5)	0.22 (0.10, 0.46)
WQS	**7.7 (1.32, 14.1)**	**2.86 (1.53, 5.37)**
Antibiotic use in past year: Yes (vs. no)	0.32 (−2.44, 3.09)	1.21 (0.91, 1.61)
Antibiotic use in past year: Unknown (vs. no)	−2.5 (−8.8, 3.85)	1.12 (0.62, 2.05)
Education: High school/associate’s degree (vs. less than high school degree)	8.12 (0.26, 16)	2.50 (1.23, 5.10)
Education: Bachelor’s degree or higher (vs. less than high school degree)	12.8 (4.45, 21.1)	3.10 (1.52, 6.30)
Race (non-white vs. white)	−8.2 (−12.4, −4.03)	*NA*
Food insecurity (insecure vs. secure)	−15.5 (−21.9, −9.18)	0.36 (0.21, 0.61)

## Data Availability

Data can be accessed upon reasonable request to data@show.wisc.edu.

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
