# Peer review of "Individual Nutrition Is Associated with Altered Gut Microbiome Composition for Adults with Food Insecurity"

_nutrients, 2022, doi:10.3390/nu14163407_

Round 1

Reviewer 1 Report

Many studies have investigated links between nutrition and the microbiome. This study investigated how food insecurity modified the association between personalized nutrition scores and variation in the adult gut microbiome in a cohort of 643 adults. There are a number of strengths to the paper, including the size of the population, the focus of food secure vs. food insecure individuals and the ecological approach to the data analysis using the Weighted Quantile Sum (WQS) regression models.  The data show that a higher My Nutrition Index (MNI) score was associated with a wider range of bacterial taxa for food insecure vs. secure, suggesting nutritional quality in food insecure individuals is important in maintaining health and reducing disparities, which has potential implications for designing interventions.

Specific Comments

1.  The use the My Nutrition Index (MNI) to generate individual nutrition scores from FFQ.  This is an interesting approach, but it is unclear how widely it is used, since all three references include authors from this manuscript.  Since the MNI compares to guidelines and seems similar to the Healthy Eating Index, which was developed specifically for alignment with the Dietary Guidelines.  Is there data on how closely the MNI and HEI align?  If that information was reported in one of their previous publications, it would be good to mention in this paper.

 2.   The MINI is a composite score of 34 dietary components. However, they only report data on the compiled indices: Macro Index, Vitamin Index, Electrolyte index and Mineral Index.  I am assuming that you can only look at the mixture indices, since the WQS 478 equation is built to have the mixture index on the predictor side of the equation. It is possible through other analyses to determine whether there are specific dietary components within the mixture index that were contributing to a greater or lesser degree to the relationship with the microbial genera (e.g., protein within the Macro Index)? This information would be important in considering the translation of the findings as well as in designing interventions.

3.  Is fiber included in the Macro Index?  It is not a major source of kcals (some due to microbial fermentation), but may modulate the microbiome.  Is alcohol in the Macro Index?

4.  The microbiome analyses appear to be based on genera alone.  Did they also determine if total OTU or indicators of microbial diversity (e.g., Shannon, Simpson) differed between food secure and insecure individuals? Even if those were not included in the WQSRS analyses, readers might be interested in this information as normative data to compare with other published data.

5. Due to inclusion of sex, age, BMI, smoking, and physical activity in the calculation of MNI, these variables were excluded as covariates or confounders in the analyses. Covariates for the My Nutrition Index included race, food insecurity, and education status. However, it is important to note that the food insecure individuals in the cohort were more likely to have a higher BMI and be a smoker. They were also more likely be non-white and less educated. Do they have data on health variables (e.g., diabetes, hypertension or metabolic syndrome) in this cohort?

6. It is interesting that several genera were significantly associated with both the MNI index and the electrolyte index. They concluded that dietary electrolytes may be driving the association seen with the overall MNI. You could conclude that higher electrolyte intake (if driven by sodium) could be a proxy for a higher consumption of processed foods.  However, the electrolyte index includes both sodium and potassium. Sodium would be higher in processed foods and breads vs. potassium in unprocessed meat, vegetables and fruit.  Therefore, it is difficult to reconcile the relationship with the electrolyte index vs. the overall MNI.  Did the FFQ collect data on sports drinks or other drinks that are high in electrolytes?

Specific Comments:

Line 91: The Dietary Guidelines for Americans are jointly issued by the U.S. Department of Health and Human Services (USDHS) and the USDA

Line 399 – Meat is not naturally higher in sodium, unless it is processed.

Reference 19:  Please update to the most current version of the DGA. 2020-2025 Dietary Guidelines report. U.S. Department of Agriculture and U.S. Department of Health and Human Services. Dietary Guidelines for Americans, 2020-2025. 9th Edition. December 2020. Available at DietaryGuidelines.gov.

Formatting issues:

ABSTRACT:  font size in lines 13-18 is smaller than the remainder of the abstract

INTRODUCTION:  move title to the following page

Table 1: move to the following page

Supplemental Table 3.  Correct spelling of Esimate

Reviewer 2 Report

The manuscript represents an interesting practical study aiming to assess the association between personalized nutrition scores, using My Nutrition Index (MNI), and the composition of the gut microbiome in adults, using 16s rRNA amplicon sequencing. The authors, also, adequately investigated whether food security modifies the association between MNI and the gut microbiome. This is novel, as little is known about this relationship. Also, the authors used a novel approach to microbiome analysis with stratified weighted quantile sum regression. By using this novel approach, the authors found a significant association between the adult gut microbiota and personalized nutrition scores (MNI), and the electrolyte index, which represents very useful finding. The method described in the manuscript has been adequately chosen and presented. This analysis is among the first to consider total dietary quality measured by a personalized nutrition scores (MNI) and specific indices on gut microbial composition. It is the first analysis to apply the stratified interaction WQS extension to microbiome data. Interpretation of the results and conclusions drawn from the results are adequate. Authors’ statements are well supported with relevant references, widely supported with their recent surveys. Discussion part is excellently structured, all the figures and tables are relevant. I find the results practical, as they may suggest nutrition is more important in shaping the gut microbiome for people who are food insecure. Possible limitations of the study and future directions have been very well addressed. I recommend publishing the manuscript in its current form. The only small change I would suggest: 371-372 vs. 376-378 - the sentences are very similar – one to be deleted.

Author Response

We would like to thank the reviewer for their review and comments on our manuscript. We appreciate the feedback, and are very pleased that you found the manuscript interesting and the findings useful. We agree that the two sentences identified were very similar, and we removed the first one.